# The Impact of Line-of-Sight and Connected Vehicle Technology on Mitigating and Preventing Crash and Near-Crash Events

**DOI:** 10.3390/s24020484

**Published:** 2024-01-12

**Authors:** Eileen Herbers, Zachary Doerzaph, Loren Stowe

**Affiliations:** 1Virginia Tech Transportation Institute, Virginia Tech, Blacksburg, VA 24060, USA; 2Department of Biomedical Engineering and Mechanics, Virginia Tech, Blacksburg, VA 24060, USA

**Keywords:** advanced driver assistance systems, naturalistic driving data, automated driving systems, connected vehicles

## Abstract

Line-of-sight (LOS) sensors developed in newer vehicles have the potential to help avoid crash and near-crash scenarios with advanced driving-assistance systems; furthermore, connected vehicle technologies (CVT) also have a promising role in advancing vehicle safety. This study used crash and near-crash events from the Second Strategic Highway Research Program Naturalistic Driving Study (SHRP2 NDS) to reconstruct crash events so that the applicable benefit of sensors in LOS systems and CVT can be compared. The benefits of CVT over LOS systems include additional reaction time before a predicted crash, as well as a lower deceleration value needed to prevent a crash. This work acts as a baseline effort to determine the potential safety benefits of CVT-enabled systems over LOS sensors alone.

## 1. Introduction

New vehicles are being equipped with a multitude of sensors to be used in advanced driver assistance systems (ADAS) and/or automated driving systems (ADS) to develop an understanding of their environment. These environmental sensors can generally be characterized as line-of-sight (LOS) sensors because they rely on information directly captured by the sensors’ field of view. However, to increase the amount of information available and to expand the sensed field of view, it is beneficial to use additional shared data from other vehicles and the infrastructure. By leveraging evolving communication systems, data shared over connected vehicle technologies (CVT) may provide a variety of performance benefits to transportation. This new level of collaborative communication has the potential to develop a collective perception of a vehicle’s environment, which could directly improve safety as events unfold.

The purpose of this research is to act as a baseline attempt to measure the potential safety impact that advanced sensors and communication methods can make in real-world crashes and near-crashes. Therefore, in this research LOS sensors are used to describe any sensors that use vision-based technology for object detection (such as cameras, RADAR, or LiDAR). These sensors are most often used in advanced driver assistance systems (ADAS) and automated driving systems (ADS). CVT may have additional sensors that are used in sharing or receiving information with other vehicles or infrastructure. Currently, the implementation of CVT is mostly found in simulation testing [1].

Previous work has characterized some of the potential advantages of LOS and CVT, such as increasing traffic speed or flow rate without any negative impact on traffic safety, improving individual mobility, providing environmental impact reduction benefits through reduced fuel use and better efficiency, and preventing/mitigating fatal and injury-causing crashes [2,3,4,5,6]. Although some of these potential advantages can be readily predicted through macrosimulation techniques, quantifying the actualized safety benefit of LOS sensors or CVT requires a more nuanced approach. This is because the specific factors leading up to a police-reported crash are generally unknown and can vary greatly between one another [7]. Generally, the prospective safety impact of more advanced vehicle-sensing technology, like LOS or CVT, is projected by estimating the number or percentage of police-reported crashes that could have been avoided if the vehicles involved were equipped with additional sensors. For example, through a meta-analysis model of 73 different studies, it was found that up to 48.07% of crashes in the US could have been prevented if all of the vehicles involved had CVT or were automated vehicles [8]. Additionally, through the analysis of GES crash records from 2005–2008, CVT or LOS sensors could conceivably have prevented 32.99% of crashes, and 47% of rear-end crashes in the US [9]. According to a preliminary study by the National Highway Traffic Administration (NHTSA), CVT could eventually prevent or mitigate about 80% of non-alcohol-related crashes [10]. These studies use crash aggregates and crash types to assume how many crashes could have been prevented if there was 100% market penetration of these technologies.

Studies like these provide a general estimate of the types of crashes that could be mitigated or avoided with the implementation of these sensors and technologies, but each crash is unique, and the actual impact of LOS and CVT may be affected by additional factors. It becomes more difficult to accurately predict the safety impact of these systems on the microscopic scale [11]. One way to do so is to evaluate the performance of current ADAS systems (which contain LOS sensors). For example, a partnership between automakers and NHTSA used real-world vehicle data from 47 million ADAS-equipped vehicles to determine that vehicles with automatic emergency braking (AEB) are 49% less likely to strike another vehicle in a rear-end crash [12]. Another way to do so is by introducing LOS or CVT sensors into simulation studies. A study in which intersection crashes were re-simulated predicted that an intersection-specific ADAS could prevent 25–59% of crashes [13]. Another simulation study using a bottom-up microscopic simulation approach to predict macroscopic statistics found that 24–87% of fatal crashes could have been avoided in scenarios involving vehicles with ADAS/ADS systems compared to fully manual driving scenarios [14]. 

In developing simulations to assess CVT effectiveness, machine learning models have been used to simulate specific events which can then be tested on the road. One study developed a long short-term memory model to predict vehicle trajectories to simulate a cut-in maneuver in a V2V environment, which was superior to traditional collision-warning models [15]. Another study developed a road safety information system using naturalistic data from connected vehicles on Korean highways to assess how connected vehicles could affect traffic safety and flow. However, this study used a macroscopic model for each section of the highway, and suggests a more microscopic calibration to assess actual crash risk [16]. Although one study used Doppler shift to assess a collision-avoidance system that specifically used only wireless communication without any LOS sensors [17], another study developed a high-level fusion of LOS sensors and wireless vehicle communication data to predict the trajectories of conflict with vehicles and pedestrians and found that this fusion enabled higher driver and pedestrian safety [18]. This fusion method is more similar to how the sensors are viewed within LOS and CVT systems in this research. However, an important piece of information that is missing from these simulations and could be beneficial in predicting the actual impact of more advanced vehicle sensors is the actual vehicle kinematic signatures before and during some of these safety-critical events (SCEs).

In this paper, a physics-based model was developed to simulate real-world crash and near-crash scenarios using naturalistic data from the Second Strategic Highway Research Program Naturalistic Driving Study (SHRP2 NDS). Naturalistic driving data provide a wealth of information before, during, and after SCEs and baseline scenarios. Especially important for this research, real-time vehicle data of near-crashes were captured, which enabled us to analyze SCEs that are not found in national crash datasets. These events were reconstructed so that the benefit of LOS sensors and CVT could be compared to the baseline scenarios that did not have either of those technological benefits. Four different crash configurations were studied, and the system activation time and resulting required deceleration to avoid these crash and near-crash events were calculated. The research in this report intends to add to the body of knowledge around the potential quantitative safety impact of vehicles equipped with LOS sensors (ADAS-equipped vehicles) and the probable added benefit of CVT systems (vehicle-to-everything (V2X)). This work was part of a larger Safety Through Disruption (Safe-D) University Transportation Center (UTC) report [19]. 

## 2. Materials and Methods

The following section includes a description of the dataset, and how events were identified to be included in the data extraction. A flowchart of this method can be found in Appendix B. The data extraction included pulling de-identified data from VTTI’s data enclave to obtain the Global Positioning System (GPS) positions of the subject vehicle to generate a map image of the location without the linked GPS coordinates. The subject vehicle in this research is used to describe the “host vehicle”, or the vehicle containing the DAS. The target vehicle is used to describe the vehicle that conflicted with the subject vehicle (i.e., it was the other vehicle in the crash or near-crash). The subject vehicles and the target vehicles were then manually tracked throughout the event to determine their trajectories and speed. This information was used to recreate the events and calculate crash-specific variables that could be used to calculate the impact that ADS technologies could have had on the outcome of these events.

### 2.1. Dataset

The Second Strategic Highway Research Program Naturalistic Driving Study (SHRP2 NDS), funded by the U.S. Federal highway Administration (FHWA), is the largest naturalistic driving study that has been undertaken to date. The SHRP2 database consists of over 5.5 million trips driven by 3542 drivers across 6 collections sites in the continental United States. These sites hosted from 150 to 450 participants each and included rural sights such as central Pennsylvania, and more populated urban areas such as Seattle, Washington. VTTI developed a data acquisition system (DAS) to support the research questions and objectives of the SHRP2 NDS program, which included compiling a dataset that could be used to support future data mining activities such as this one [20]. SHRP2 was used because of the availability of a “breadcrumb” trail of the GPS location, speed, acceleration, and other timeseries data [21]. The DAS facilitated the collection of the following data of interest to this study:• video data of the forward view;• subject vehicle speed;• subject vehicle yaw and yaw rate;• GPS latitude and longitude.

These variables were chosen because they could be used to recreate events of interest within a simple physics-based model. However, the DAS collected a variety of additional variables such as multiple video angles, machine vision, accelerometer data in all 3 axes, driver cell phone use, vehicle network data, and more. The study was conducted in accordance with the Declaration of Helsinki and approved by the Institutional Review Board of Virginia Tech (IRB #18-957 23 October 2018).

### 2.2. Event Identification

To correctly compare the potential benefit of CVT over LOS sensors, a subset of the crash and near-crash events from SHRP2 were identified. These events included ones where the view of target vehicle was obstructed so that the LOS sensor would not be able to perceive the target vehicle prior to an imminent potential conflict. Therefore, the capability of an LOS system would be limited, while a CVT system could provide a benefit to the operation of the associated safety system. Therefore, the conflict object, or target vehicle, was out of sight for the majority of the time leading up to the event. Figure 1 illustrates an example of such an event. Although this constitutes a strong selection bias, these specific events were chosen because they showed the most promise to fulfill the purpose of determining how the addition of LOS and CVT systems could mitigate or prevent real-world crash and near-crash scenarios.

From initial data mining, 594 events showed initial promise to be relevant to this research because there was a view-obstructing object involved within the incident and the necessary data elements were available. These events were defined by the SHRP2 dataset and included crashes and near-crashes. A crash was defined as “any contact that the subject vehicle has with an object” and a near-crash was defined as “any circumstance that requires a rapid evasive maneuver by the subject vehicle, or any other vehicle, pedestrian, cyclist, or animal, to avoid a crash” [20].

The candidacy of each event was then rated by its relevancy to the project and its ability to be reconstructed; those that were not good candidates included events with insufficient video, unpredictable conflict object maneuvers, host driver error, and more. Insufficient video included video with heavy precipitation, video with insufficient lighting, or video that was unavailable. Unpredictable conflict object maneuvers included any vehicles or animals that made erratic movements that would be difficult to reproduce with a simple physics-based model. Thus, 18 crashes and 162 near-crashes were identified that promised a strong ability to be recreated for the purposes of this project. 

After these events were determined, they were reviewed once again to classify the leading cause of conflict. Table 1 provides a summary of the obstruction type for the events with strong candidacy. 

### 2.3. Data Extraction

To obtain a better picture of the environment during a crash or near-crash event, a birds-eye view of the location of the event was created. The corresponding Google map image was extracted for each event and overlayed with the relative location of the subject vehicle. To keep the locations anonymous, the actual GPS coordinates of the subject vehicle were kept in the VTTI data enclave and converted to pixel locations, relative to the map image. The other data taken from SHRP2 were the instantaneous kinematic details of the subject vehicle and the front-facing video of the subject vehicle. Together, these four pieces of data were used to reconstruct the events in a physics-based model.

### 2.4. Event Reconstruction

#### 2.4.1. Identifying Important Timepoints

The first step in recreating the event was to superimpose the subject vehicle trajectory over the map image. The accuracy of automotive-grade GPS is not always good enough to directly overlay on the map. For example, in one case, the subject vehicle appeared to be offset from the road and driving in the grass a few meters to the side. Therefore, the relative kinematic information collected from the DAS was used to generate the trajectory given a set of initial conditions. Points were marked individually on the map for the subject vehicle’s initial location and the kinematic information from the subject vehicle was used to recreate its trajectory.

A graphical user interface (GUI) was developed to complete the following two tasks:Determining the impact proximity frame (timestamp) by watching the event video. This is the approximate timestamp in which the subject vehicle and target vehicle come into contact (or near contact for near-crash events). This is later referred to as the conflict time.Then, two frames (timestamps) are identified within the video that correspond to two locations of the subject vehicle on the map. The frames are chosen based on the ability to accurately place the concurrent subject vehicle position and heading on the map (i.e., lane markings, buildings, trees, etc.).

#### 2.4.2. Calculating Subject Vehicle Trajectory

Once two positions, headings, and corresponding timestamps were identified, the vehicle’s trajectory throughout the event was calculated given the vehicle kinematic data that were extracted previously from the DAS. This was performed by using the vehicle’s starting position (*x_n_*), heading (*θ_n_*), and speed (*v_n_*) in a basic iterative trajectory formula shown in Equation (1).
(1)$$x_{n+1}=x_{n}+v_{n}\cos\theta_{n}$$

This produced a photo of the event trajectory superimposed on the corresponding map as shown in Figure 2. These photos were reviewed to determine if it would be beneficial to modify and repeat earlier steps for any events that had an unexpected trajectory. If the trajectory was unexpected (Figure 2, right), the associated trace factor and theta shifter were internally developed to determine more accurate positions and headings for the subject vehicle in the first step. The trace factor is the ratio between the trajectory distance of the event (as determined from the DAS) and the distance calculated from the vehicle positions chosen during the video review. The theta shifter is the difference in heading (degree). 

#### 2.4.3. Determining Locations of Objects of Interest

The GUI generated a .MAT file that contained the positions of the subject vehicle, view-obstructing objects, and the target vehicle (or conflict object). To generate the locations of the objects of interest, the subject vehicle front camera and the position of the subject vehicle on the map were displayed at corresponding timestamps.

First, the impact proximity frame (conflict time) was brought up and the location of the target vehicle was identified on the map. Then, the event was reversed by 4 frames at a time until the frame in which the target vehicle was no longer visible was reached. At this frame, each object that could be obstructing the view of the subject vehicle was identified and its location was marked. Then, the event continued 4 frames at a time and the corresponding locations of each view obstructing object and the target vehicle was marked. These locations were marked up until the conflict time. 

This step went quickly if the position and heading of the subject vehicle was accurate (as determined by the previous two steps), there was only one target vehicle, and only one stationary view-obstructing object. This step took considerable effort if there were many view-obstructing objects that move throughout each frame, there were multiple target vehicles, or the subject vehicle position was not accurate. 

### 2.5. Physics-Based Model

Each event now had the relative locations and headings on the subject vehicle, the target vehicle(s), and the view-obstructing object(s) at certain time frames of the event. Linear interpolation was used to fill in the locations of these objects for the missing timestamps. This allowed for different parameters to be manipulated to simulate different scenarios that stem from one event. For the purposes of this project, three scenarios were simulated. The first scenario acts as the base case, which used the data in the original event reconstruction. The second scenario acts as though the subject vehicle has line-of-sight (LOS) technology. For simplicity in calculation, the subject vehicle detects the target vehicle when an uninterrupted line can be drawn from the centroid of each vehicle as shown in Figure 3. Since LOS sensors differ in range and width, this simplification allows for consistent calculation. We expect that most sensors will need to view much of the vehicle in order to correctly detect it, and this was a simple way to exemplify this expectation. The third scenario describes when the subject vehicle and target vehicle are using CVT (i.e., the subject vehicle knows the location, speed, acceleration, and trajectory of the target vehicle). These assumptions were used to calculate how environmental sensors (LOS) and information-sharing between traffic participants and smart infrastructures (CVT) could impact vehicle safety in these crash and near-crash scenarios. Using these data, two pieces of information were calculated for each scenario: (1) the activation time in which a potential conflict is identified in both CVT and LOS ADS and (2) the required deceleration of the subject vehicle to prevent a crash in both CVT and LOS ADS.

#### 2.5.1. Activation Time before Conflict Calculation

To calculate these two pieces of information, the relative speed, heading, and locations of the target vehicle and view-obstructing objects were needed. The instantaneous speed, v_n_, was calculated by taking a simple derivative of the pixel location of each object over time. The pixel distance was converted to distance in meters by using the zoom factor used to produce the map image (i.e., the zoom factor, zf, was a fraction that correlated the number of pixels to a distance in meters).

Then, in each frame, the current speed, location, and heading of the subject vehicle and target vehicle were used to determine the expected trajectory of each vehicle (Equation (1)). A conflict was identified if the vehicle centroids came within 4 m of each other at some point within their predicted trajectories. 

The first time a potential conflict was identified in the data, defined when the CVT system would activate. The LOS activation timepoint would occur once the subject vehicle could “see” the target vehicle (as shown in Figure 3). Therefore, a LOS system would activate either concurrently, or after a CVT system. By taking the difference between those two vectors, it could be determined how much earlier a CVT system could notify a driver of a potential conflict over an LOS system. The system activation time before the conflict time is essentially the commonly used safety surrogate measure, time-to-collision (TTC). However, because the real-world conflict time was known, we used this predicted system activation time relative to the actual conflict time.

#### 2.5.2. Required Deceleration

With the distance between each vehicle (*d*), the predicted point of collision, the time until collision (*t*), and the speed of both vehicles (*v_S_* and *v_T_*), the minimum required deceleration (*−a*) to prevent a crash can be calculated by Equation (2).
(2)$$-a=\frac{d-\left|v_{S}-v_{T}\right|t}{t^{2}}$$

The collision avoidance strategy assumes that the subject vehicle does not swerve and the conflict is avoided with braking only; the driver (or vehicle) does not have to perform any other evasive maneuvers. Additionally, a simplifying assumption is made such that the target vehicle does not accelerate or swerve once a conflict is identified. 

## 3. Results and Discussion

As discussed earlier in Section 2.2 Event Identification, 18 crashes and 162 near-crash events were reconstructed. After analyzing these 180 events, 68 events still possessed with usable data. Furthermore, 112 events were excluded due to incorrect satellite images (e.g., major construction since the date of the event, incorrect GPS data points in SHRP2) or missing kinematic data within the time of interest for our project. The resulting 68 events were separated into four crash configuration categories, and the following two values were calculated: (1) the difference between CVT and LOS activation time, and (2) the minimum required deceleration to avoid a collision. 

### 3.1. Crash Configuration Categories

To organize the events and find significance within the values calculated, events were categorized into four crash configurations. These four categories were based off the General Estimates System (GES) accident type diagram and can be found in Appendix A, with the number of events within each configuration shown in Table 2.

Even though rear-end collisions are the most frequent type of crash in the US [22], the study did not include a proportional number of these events. This is because they are generally not caused by a visual obstruction, but rather due to driver distraction or lack of driver awareness. Although the events in this research make up a relatively low sample size for each crash configuration, it is important to note that the left turn across path configurations made up a significant number of crash and near-crash events that could be mitigated by LOS and CVT systems within this sample. Therefore, it could be beneficial to focus on left turn across path scenarios in future work involved with assessing the safety of technologically advanced vehicles.

### 3.2. Activation Time before Conflict

The activation time represents the amount of time between when each system detected an imminent conflict and the actual time of conflict. This concept somewhat represents the common safety surrogate measure, time-to-collision (TTC), but is calculated slightly differently here since the time of the actual conflict is known. Figure 4 shows the difference in activation time between CVT and LOS sensors. Each dot represents the actual value for each scenario, the X represents the mean value, the horizontal line represents the median value, the box encompasses the first and third quartile, and the whiskers extend to the maximum value that is within 1.5 times the inner quartile range. 

Across all scenarios, the CVT system activated 0.51 ± 0.15 s before a LOS sensor detects the target vehicle on average. This means that CVT could provide an additional ½ second of reaction time over LOS systems. Additionally, since all crash and near-crash events are generally classified together as safety-critical events (SCEs), an additional ½ second could allow for earlier activation of forward collision-warning systems or automated emergency braking in these scenarios [20]. 

### 3.3. Required Deceleration

The required deceleration is a value calculated to determine the minimum deceleration necessary to avoid a crash or near-crash if the vehicle began braking at the system activation time. Figure 5 is a box and whisker plot of the deceleration values required to prevent a potential conflict. Similar to Figure 4, the X represents the mean value, the horizontal line represents the median value, the box encompasses the first and third quartile, the whiskers extend to the maximum value that is within 1.5 times the inner quartile range, and each dot represents the actual value. 

Across all four crash configuration categories, the average required deceleration of CVT systems vs. the LOS systems were 3.79 m/s^2^ and 6.22 m/s^2^, respectively. Generally, a deceleration value over 0.45 g (4.41 m/s^2^) is considered hard braking [23]. Therefore, CVT systems could reduce the need for hard braking, and reduce the average deceleration required by 2.43 m/s^2^ to avoid the conflict by braking alone.

Some events required excessive deceleration values. Figure 6 shows the number of events binned by their respective required deceleration to avoid a conflict (anything above 14 m/s^2^ is shown as 14+). In general, a deceleration value of less than 1 g (9.8 m/s^2^) is reasonable for most modern light vehicles equipped with crash-avoidance systems [24]. This figure shows that more LOS events (than CVT events) require an acceleration value of more than 1g as depicted by the black dashed line. Additionally, 91.2% of the CVT events and 75.0% of the LOS events analyzed required a deceleration value less than 1g, implying that a vehicle equipped with LOS features alone could prevent 75.0% of conflicts within this dataset, while a connected vehicle could prevent 91.2% of conflicts. 

## 4. Conclusions

In this research, crash and near-crash scenarios from the Second Strategic Highway Research Program Naturalistic Driving Study (SHRP2 NDS) were simulated via a physics-based model to calculate the potential safety benefit of line-of-sight (LOS) sensors and connected vehicle technology (CVT). Previous work has predicated the potential safety impacts of LOS and CVT through an estimation of the types of police-recorded crashes that could have been avoided if these sensors and systems were in place, or by simulating different crash scenarios with these technologies in place. A missing piece has been the use of real-time kinematic data of vehicles during a crash, as well as using other safety-critical events, such as near-crashes, to analyze how additional LOS or CVT could perform in these scenarios. This research acts as a baseline attempt to measure the potential safety impact that advanced sensors and communication methods can provide in real-world safety-critical events (SCEs).

This project looked at four different crash configurations, including left turn across path, rear-end, perpendicular, and turn into same direction. The turn across left path configurations contained the largest number of crash and near-crash scenarios that could be addressed through LOS sensors or CVT systems. Therefore, it could be beneficial to focus more attention on the accuracy of these sensors specifically for left-turn maneuvers.

On average, the CVT system would activate 0.51 ± 0.15 s before a LOS sensor detects the target vehicle. This means that CVT could provide about an additional half-second of reaction time over LOS systems. In future research, determining how this calculated value might change at different speeds could greatly affect the added safety benefit of some of these sensors. On average, the required deceleration of CVT systems vs. the LOS systems to avoid a conflict were 3.79 m/s^2^ and 6.22 m/s^2^, respectively. Additionally, 91.2% of the CVT events and 75.0% of the LOS events analyzed required a deceleration value less than 1 g. 

From the required deceleration interpretation, it is expected that any event which required a deceleration value greater than 9.8 m/s^2^ could have resulted in a crash. However, only 4.4% of the actual events that were analyzed resulted in a collision; the remaining 95.6% were near-crashes. This is because one or more of the vehicles involved performed an evasive maneuver in addition to braking, which is often how near-crashes are categorized [25]. The simulated event then calculated a required deceleration that was higher than the baseline event to avoid the collision because it did not include a swerving maneuver. These near-crashes were used for this research to increase the sample size and they can be used as a potential surrogate measure to crashes in similar scenarios [26]. Although the near-crashes did not result in an actual police-reported crash, these are important to use in determining the potential safety impact of automated driving systems since these were events in which the driver performed a successful evasive maneuver. However, since the results from the simulated LOS and CVT systems only included braking as an evasive maneuver, further research could look into how swerving could be used to avoid some of these conflicts. This would be especially beneficial for ADS development. Most of the drivers in these near-crash events were able to avoid the crash with a combination of swerving and braking, so it would also be beneficial to see if vehicles with additional sensors and more advanced driving assistance systems could do the same.

Although this research includes only a small sample of SCEs, this work demonstrates how certain safety-surrogate measures can be used to measure the potential safety impact of more advanced sensors and communication methods. It would also be beneficial to calculate these same surrogate measures with a larger dataset for use in different baseline scenarios without a visual obstruction to compare the results. The events analyzed were specifically chosen because CVT is most likely to have an impact in scenarios in which LOS sensors are blocked. However, many SCEs occur when there are no visual obstructions, and CVT has the potential for also avoiding or mitigating these events.

Finally, these simulations assumed that the sensors would have 100% accuracy in determining an imminent conflict. More conservative estimates could be made to account for sensor inaccuracies or additional reaction time within technological systems. This research acts as a baseline sample of how to use real SCEs to predict the potential safety benefit of advanced vehicle sensors, and which events should be focused on for future research.

## Figures and Tables

**Figure 1 sensors-24-00484-f001:**
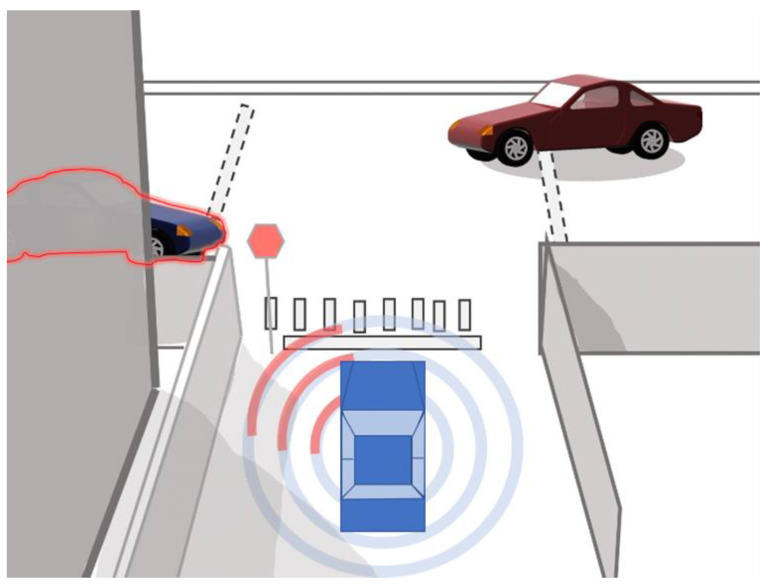
Example of a visual obstruction event.

**Figure 2 sensors-24-00484-f002:**
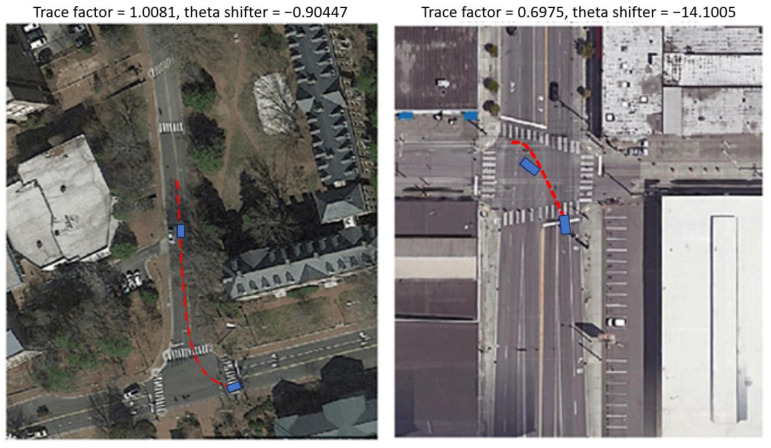
Review step showing an example of correctly calculated trajectory (**left**) and incorrectly calculated trajectory (**right**).

**Figure 3 sensors-24-00484-f003:**
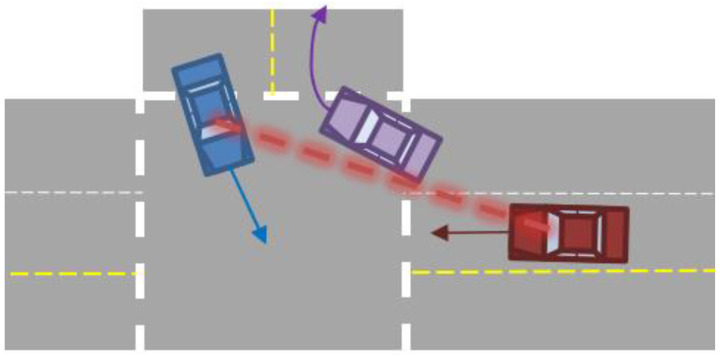
The centroid of the target vehicle (blue) is within the line of sight of the subject vehicle (red) around the visual obstruction object (purple), which determines LOS activation.

**Figure 4 sensors-24-00484-f004:**
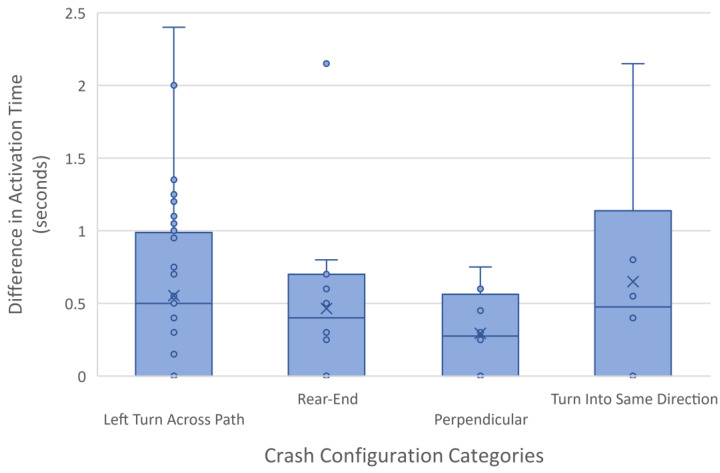
Box and whisker plot of the difference in activation time (seconds) between LOS and CVT systems separated by crash configuration categories. Each circle represents the actual value for each scenario, the X represents the mean value, the horizontal line represents the median value, the box encompasses the first and third quartile, and the whiskers extend to the maximum value that is within 1.5 times the inner quartile range.

**Figure 5 sensors-24-00484-f005:**
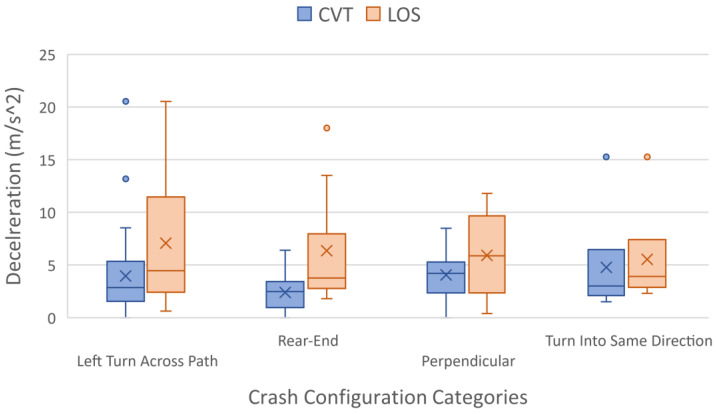
Box and whisker plot of mean required deceleration (negative acceleration) separated by crash configuration category and sensor type. The X represents the mean value, the horizontal line represents the median value, the box encompasses the first and third quartile, the whiskers extend to the maximum value that is within 1.5 times the inner quartile range, and the circles represents any values outside of the whisker range.

**Figure 6 sensors-24-00484-f006:**
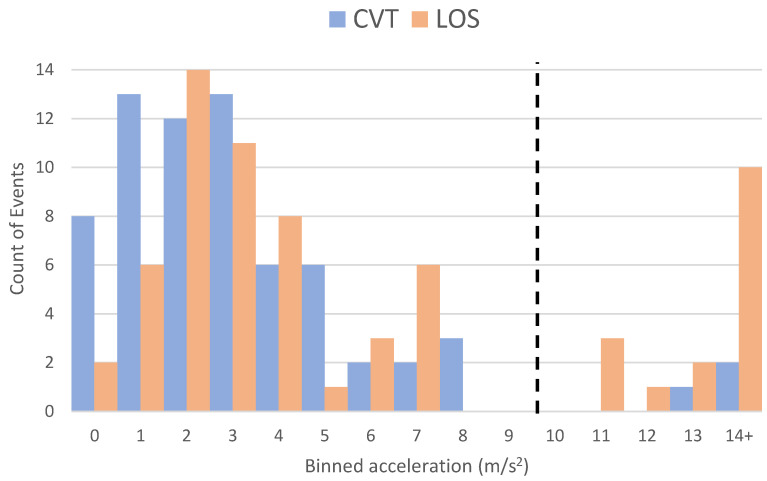
Count of events binned by required acceleration to avoid the crash or near-crash separated by sensor type.

**Table 1 sensors-24-00484-t001:** Events with strong candidacy by obstruction type.

Obstruction Type	Near Crash	Crashes
vehicle	126	11
bend in path (lateral or vertical)	9	1
none (small agent)	0	1
fog	0	1
other (e.g., building, vegetation)	27	4

**Table 2 sensors-24-00484-t002:** Crash configuration categories and the number of events analyzed.

Crash Configuration	Number of Events
left turn across path	40
perpendicular	8
rear-end	14
turn into same direction	6

## Data Availability

Data used for this research is available on VTTI’s Dataverse here: https://safed.vtti.vt.edu/projects/impacts-of-connected-vehicle-technology-on-automated-vehicle-safety/. (accessed on 11 January 2024).

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
