# Peer review of "The Impact of Line-of-Sight and Connected Vehicle Technology on Mitigating and Preventing Crash and Near-Crash Events"

_sensors, 2024, doi:10.3390/s24020484_

Round 1

Reviewer 1 Report

Comments and Suggestions for Authors

The paper is very interesting, they use the sensor technology for the road safety, including line-of-sight (LOS) and connected vehicle technologies (CVT). The result of the study can be implemented in the real-world. In term of technique the author has well-writing. the observation is based on the  Declaration of Helsinki. However, in term of the reassert paper, there are some points must be clearly explained for the benefit of the readers. My comments are follows:

1. In introduction section, the examination of the LOS and CVT are very short the author must explain more detail, for example how the sensor work? What is the previous research? How about the using in the car in currently.

2. The contribution of work is needed in explain in the introduction.

3. The references in the paper are very old. It must be up to date.

4. The procedure of the experiment quite complex. So the infographic or flow chart is needed, to make the reader understand easier.

Reviewer 2 Report

Comments and Suggestions for Authors

The paper The Impact of Line of Sight and Connected Vehicle Technology on Mitigating and Preventing Crash and Near-Crash 3 Events deals with a topic related to traffic safety in relation to vehicles equipped with driver assistance systems.

Overall, the article is well organized and clearly presents the concept of research that involves comparing the effectiveness of the use of two driver assistance systems in preventing road accidents. And this is where the first doubt arises - do the authors try to assess and compare the potential of both systems or do they assume that one of them (CVT) is better than the other (LOS) (line 125)? This assumption seems to slightly contradict the title of the article.

The Authors should provide some details about research area – were the crashes recorded in or outside an urban area.

In formula 2, the symbol of deceleration should be rather expressed as “d” than “a”, “a” normally stands for acceleration. What was the vehicles’ speed that authors used in calculations?

The results are interesting, but some doubts are raised by the small number of analyzed data especially left turns across path and turns into same direction. So, it would be worth to extend analysed data, having access to a wide database, to obtain a more complete analysis.

Although the authors explain the method of event selection (Section 2.2), the article lacks a similar comparison made for events occurring in good conditions without additional visibility restrictions (given in Fig. 1) - after all, lots of accidents happen in the area of intersections with good visibility.

Finally, it is worth clearly emphasizing what the purpose of the article is, is it only to compare both driver assistance systems or to indicate one of them as the leading one regardless various road conditions? And therefore for implementation in new cars. If the goal is only to compare both systems, then what are the practical results of the results obtained - this is missing in the final conclusions.
